# Comorbidities and Complications in Idiopathic Pulmonary Fibrosis

**DOI:** 10.3390/medsci6030071

**Published:** 2018-08-30

**Authors:** Esteban Cano-Jiménez, Fernanda Hernández González, Guadalupe Bermudo Peloche

**Affiliations:** 1Hospital Universitario Lucus Augusti, Respiratory Department, 27002 Lugo, Spain; 2Hospital Clínic, Respiratory Department, 08036 Barcelona, Spain; hernandez.gonzalez.fer@gmail.com (F.H.G.); lupebermudope@gmail.com (G.B.P.)

**Keywords:** idiopathic pulmonary fibrosis, comorbidities, complications

## Abstract

Though idiopathic pulmonary fibrosis (IPF) is characterized by single-organ involvement, many comorbid conditions occur within other organ systems. Patients with IPF may present during evolution different complications and comorbidities that influence the prognosis and modify the natural course of their disease. In this chapter, we highlight common comorbid conditions encountered in IPF, discuss disease-specific diagnostic modalities, and review the current treatment data for several key comorbidities. The diagnosis and treatment of these comorbidities is a challenge for the pulmonologist specialized in interstitial lung diseases (ILDs). We will focus on pulmonary emphysema, lung cancer, gastroesophageal reflux, pulmonary hypertension, obstructive sleep apnea (sleep disorders), and acute exacerbation of IPF.

## 1. Introduction

Idiopathic pulmonary fibrosis (IPF) is a progressive, chronic, and unpredictable disease with few useful treatments and a poor survival.

Although the appearance of new treatments, such as nintedanib and pirfenidone, has given new hope to these patients, its management is complex and is influenced by the coexistence of several comorbidities. The presence of these comorbidities may decrease patients’ survival and quality of life and might accelerate the progression of the disease. Some comorbidities might coexist with IPF because of a common risk factor for both diseases (e.g., tobacco and lung cancer) or, in other cases, they could be a consequence of IPF itself (e.g., acute exacerbation or pulmonary hypertension). 

There is increasing evidence that the early diagnosis and treatment of comorbidities is as important as the treatment of idiopathic pulmonary fibrosis itself.

## 2. Pulmonary Emphysema 

Smoking is a common risk factor for both emphysema and pulmonary fibrosis [1,2,3]. Not surprisingly, about 30% of IPF patients have concurrent pulmonary emphysema, including 8–27% with ≥10% emphysematous involvement throughout the lungs [4]. 

The syndrome of combined pulmonary fibrosis and emphysema (CPFE) has been proposed as an important phenotype of pulmonary fibrosis and is defined by the presence of emphysema and parenchymal pulmonary fibrosis in the same patient [1]. In this proposal, CPFE includes pulmonary fibrosis other than IPF and is considered an independent entity, differentiated from IPF [3], but with similar radiological, epidemiological, clinical, and functional characteristics.

It has been suggested that cigarette smoke is the main causal factor, given the fact that a history of smoking is constant in all the published cohorts [2]. Several common mechanisms have been described in the physiology of emphysema and idiopathic pulmonary fibrosis such as: (a) aging cell mediated by telomerase shortening, (b) dysregulation of the molecular pathways that respond to mechanical stress, and (c) alterations in the production of modulating proteins of the cell cycle in fibroblasts (caveolins).

Kaolinite or aluminium silicate, for example, is an inorganic substance with industrial use which is present in tobacco smoke and has been isolated in the alveolar macrophages of smokers with pulmonary fibrosis and emphysema [1,3,5].

Usually, individuals with CPFE tend to be males with an extensive smoking history and increased oxygen requirement [1,2,3,4]. 

The coexistence of emphysema and fibrosis determines a characteristic functional profile that contrasts with the degree of dyspnea manifested by these patients. 

Pulmonary mechanics are altered by emphysema and pulmonary fibrosis. In emphysema, elastic forces are reduced, and both lung compliance and pulmonary volumes are increased. Pulmonary fibrosis leads to increased lung elastic recoil, decreased compliance, and reduced lung volumes [3]. The simultaneous presence of these alterations with opposed mechanics can explain the “pseudonormalization” of lung volumes in patients with CFPE.

The forced vital capacity (FVC), the forced expiratory volume in the first second (FEV_1_), and the total pulmonary capacity (TLC) are usually within normal values or only slightly decreased, while the diffusing capacity of carbon monoxide (DLCO) is disproportionately reduced (Figure 1) [6]. There is also an important exercise-induced desaturation during the six min walking test.

These physiologic hallmarks of CPFE likely reflect the opposing impact of parenchymal fibrosis and parenchymal destruction on airflow and lung volumes, along with their additive impact on gas exchange. High-resolution computed tomography (HRCT) is part of the routine diagnostic evaluation of all patients with suspected IPF [4], and routine semi-quantitative assessment of emphysematous involvement may help to easily identify those with CPFE, once a multidisciplinary diagnosis has been established (Figure 2).

The recognition of CPFE has potential management implications. Some studies suggest that CPFE is associated with reduced survival [7], but others have not replicated this observation [2,7]. Paradoxically, patients with CPFE have a slow rate of FVC decline, perhaps due to the impact of emphysema on the manner in which FVC reflects progressive fibrosis [8]. An increased prevalence of pulmonary hypertension (PH) has also been demonstrated among those with CPFE [2,7], which may also impact survival [2,8] and is the principal negative prognostic factor for this condition.

Regarding therapy, nintedanib and pirfenidone seem to have a similar effect in CPFE patients as that produced in subjects with IPF alone. The ASCEND (Pirfenidone) and INPULSIS (Nintedanib) studies recruited patients with emphysema. However, the whole spectrum of severity of emphysema was not represented in those studies, because those patients who presented an obstructive ventilatory pattern or a very severe reduction of DLCO on lung function testing were excluded. It is unclear whether patients with IPF and CPFE can benefit from treatments with inhaled drugs, such as long-acting beta-agonists, long-acting muscarinic antagonists, and inhaled corticosteroids [2,6,7,8]. We think that clinicians should consider the addition of these therapies, according to chronic obstructive pulmonary disease consensus guidelines [8], but this is an empiric non-evidence-based recommendation. The recommendations for smoking cessation counseling and for prescribing long-term oxygen therapy, pulmonary rehabilitation, and vaccination in patients with CPFE are based on the evidence available for patients with chronic obstructive pulmonary disease (COPD). Therefore, the endorsement of these actions in this group of patients is also empirical.

## 3. Gastroesophageal Reflux 

Gastroesophageal reflux (GER) is another common comorbidity in patients with IPF [9,10,11], and both conditions can simultaneously be present in 60% of cases. The prevalence might be even higher, because some studies that used esophageal pH monitoring suggest that GER may affect over 80% of individuals with IPF [12,13]. 

If GER is severe, it may imply pulmonary aspiration of gastric fluid, which may have a role in the natural history of IPF, as it might be a trigger of alveolar damage.

Esophageal pH monitoring remains the gold standard for diagnosis of acid GER, with a reported sensitivity and specificity over 80% [14,15]. Recent studies suggest that multichannel intraluminal esophageal impedance testing may be a better modality for detecting both acid and non-acid GER, but this test is not widely available at present [15,16].

The survival of IPF patients might be reduced if GER is left untreated. Medications to treat GER are a predictor of better survival in subjects with IPF. Reflux has also been related to exacerbations in patients with pulmonary fibrosis. It has been demonstrated that bronchoalveolar lavage (BAL) pepsin levels are raised in 30% of the cases with acute exacerbation of IPF (AE-IPF) [17]. These findings support the hypothesis that occult aspiration of gastric content plays a role in some patients with AE-IPF.

On the other hand, it has been found that the presence of reflux occurs more frequently when pulmonary fibrosis is asymmetric, with one lung more affected than the other, and this finding suggests a relationship between these diseases [18].

In patients in whom both diseases occur, reflux is often silent and associated with preserved esophageal function but with slow clearance of gastric acid, and it is usually present in the supine position. 

There is a need for further investigation of the association between GER and IPF and the effect of anti-acid therapy in patients with this disease. Although the effectiveness of anti-reflux surgery is controversial, and, thus, this treatment should not be performed routinely, it should be considered for selected IPF patients.

## 4. Lung Cancer

Compared to the general population, individuals with IPF have a nearly five-fold increased risk of developing lung cancer, with 3–22% of cases affected, and an estimated incidence of 11 cases per 100,000 person-year [19,20,21].

The strong link between IPF and cigarette smoking [4] likely explains a portion of the increased lung cancer risk, as the overwhelming majority of patients with IPF who develop lung cancer have a history of tobacco consumption.

Some studies show that squamous cell carcinoma predominate over adenocarcinoma [22], while a recent investigation of IPF-related adenocarcinoma demonstrated a high frequency of bronchiole-associated markers in IPF cases compared to non-IPF controls, suggesting that these tumors may arise from abnormally proliferating bronchioles in areas of honeycomb cyst [23].

The survival among those patients with IPF and comorbid lung cancer is poor [22,23], and death is often related to clinical deterioration caused by the malignancy, as similar rates of pulmonary function decline have been demonstrated in those subjects with and without comorbid lung cancer [22].

While surgical resection of early-stage lung cancer may be curative, IPF severity and disease evolution must be taken into consideration, because of the increased risk of postoperative morbidity and mortality. These patients have a higher mortality associated with the surgical treatment, usually due to the development of an acute exacerbation (5–15%), which leads to a short-term mortality of approximately 50%. Exacerbations may also occur after treatment with radiation therapy or chemotherapy (especially in regimens that include docetaxel) [22]. The strategies used to improve the surgical prognosis of these patients include performing sublobar lung resections, reducing fluid overload, avoiding pulmonary hyperinflation, employing high-flow oxygen therapy during surgery, and using prophylactic antibiotic treatment [23].

Recent studies suggest that the anti-proliferative effects of pirfenidone and nintedanib may synergize with concurrent chemotherapeutic oncologic agents, but additional research is needed to confirm this hypothesis [22,23,24,25].

## 5. Pulmonary Hypertension

Precapillary PH is defined by a mean pulmonary arterial pressure (mPAP) ≥25 mmHg, a pulmonary artery wedge pressure ≤15 mmHg, and elevated pulmonary vascular resistance >3 wood units with a cardiac index <2.5 L/min/m^2^ (Table 1) [26]. An important complication that is strongly linked to the increased morbidity and mortality in patients with IPF is the presence of PH. Pulmonary hypertension is more frequently found at advanced stages of the disease or when emphysema is associated, as in the combined pulmonary fibrosis and emphysema syndrome [27,28,29]. The actual prevalence of PH in patients with IPF is difficult to establish. Most of the studies on this topic are case reports and retrospective series. In early stages of the disease or at diagnosis, 8% to 15% of IPF patients may already have PH [29,30]. However, as IPF advances, this frequency rises to 32–50% of patients [31]. Subsequent studies have largely supported that this proportion of PH may be increased by comorbidities such as obstructive sleep apnea, cardiac diastolic dysfunction, or pulmonary thromboembolism [32].

Pulmonary hypertension in patients with IPF is a challenging clinical diagnosis, because the symptoms are very similar for both entities. PH should be suspected in IPF patients with dyspnea on exertion, oxygen requirements that are out of proportion to pulmonary function impairment, severe limitation to exercise capacity, DLCO values that are disproportionately decreased in relation to the values of spirometry, evidence of right heart failure on physical exam, and evidence of pulmonary artery enlargement or right ventricular hypertrophy on imaging studies [33,34,35]. In addition, it has been observed that there are differing phenotypes of PH in IPF. It is recognized that there are patients with IPF who develop PH as a consequence of extensive, underlying fibrosis, whereas a smaller subset of patients is found to have severe PH in the setting of mild to moderate fibrosis [36].

It has already been shown that antifibrotic treatments can have a positive effect on the disease course of patients with PH and IPF. However, given that IPF is a progressive disease, efforts are still to be made to detect PH as early as possible. The follow-up for PH assessment in patients with IPF is also controversial, because there is not a validated screening tool. Transthoracic echocardiography (TTE), which estimates the right ventricular systolic pressure (RVSP) by the systolic tricuspid regurgitation velocity at Doppler as a surrogate for mPAP, is the first and most widely used test to search for PH [35]. Although TTE-estimated RVSP > 35 mmHg has a high sensitivity for detecting PH in patients with IPF, the specificity of this RVSP cut-off value is only 29% [6,37,38]. Because TTE has such a low accuracy in patients with IPF, right heart catheterization (RHC) remains the gold standard for PH diagnostic confirmation, but it is not appropriate as a routine screening tool in clinical practice because of its invasiveness [34,39]. Therefore, efforts have being made to identify less invasive and more accessible diagnostic tools for PH screening, such as scoring systems based on echocardiographic signs of right heart dysfunction, pulmonary artery diameter to ascending aorta diameter ratio (PA/Pa) on chest computed tomography, pulmonary physiology tests, biomarkers, or a combination of multiple techniques. Fukurama et al. have developed a scoring system which is based on three variables: DLCO < 50% of the predictive value, PA/Pa ratio ≥ 0.9, and PaO_2_ < 80 mmHg [40]. When the three criteria are present (score of 3), precapillary PH is likely to be confirmed with a specificity of 95.8% and a negative predictive value of 85.1%. Precapillary PH is unlikely in patients with a score of zero, and as a consequence, RHC should not be needed to rule out an elevated mPAP in a significant proportion of patients [40]. Unfortunately, these tools have not been validated yet. Additionally, Kimura et al. reported that mPAP > 20 mmHg at the initial evaluation of patients with mild-to-moderate IPF was a better cut-off point for survival [35]. Therefore, many studies have suggested the importance of an early and accurate diagnosis of precapillary PH among patients with IPF for predicting prognosis.

In addition to enhance the non-invasive detection of PH, there is a need to evaluate new treatments that might improve either the hemodynamic or the clinical parameters in patients with IPF-PH. Therapies used to treat group 1 pulmonary arterial hypertension, such as endothelin receptor antagonists, phosphodiesterase-5 inhibitors, guanylate cyclase stimulators, or prostacyclin analogues, have not been effective to treat the underlying PH in the setting of IPF. Raghu et al. reported negative results for ambrisentan, an endothelin receptor type-A selective antagonist, for the treatment of IPF in patients with RHC-proven PH [41]. The trial was finished early because of lack of benefit in the subgroup of IPF patients with underlying PH. A small trial with riociguat, a guanylate cyclase stimulator, was also prematurely stopped because of an increased risk of death and other serious adverse events in the active treatment arm. STEP-IPF (Sildenafil Trial of Exercise Performance in Idiopathic Pulmonary Fibrosis) was a randomized controlled trial on the use of sildenafil, a phosphodiesterase-5 inhibitor, in patients with advanced IPF (defined by a baseline DLCO < 35% of the predictive value). Although the primary endpoint of a significant increase of the six min walking test (6 MWT) distance was not reached, improvements of dyspnea score, quality of life, oxygenation, and DLCO were noted among patients treated with sildenafil [42]. Hoeper et al. have analyzed COMPERA (Comparative, Prospective Registry of Newly Initiated Therapies for PH) data for patients with PH and idiopathic interstitial pneumonia [43]. Eighty percent of the patients in this registry had been treated with phosphodiesterase type 5 inhibitors. The accepted criteria for a positive therapeutic response to PH therapies were either an increase of at least 20 m in the 6 MWT distance or an improvement in functional class. Those patients who reached any of these criteria tended to have better survival. However, there is no evidence up to date of a positive effect on outcomes in those patients with PH secondary to interstitial lung diseases treated with PH therapies. Based on these data, the role of sildenafil as add-on therapy in combination with anti-fibrotic treatments for patients with PH in the setting of IPF is being investigated in several clinical trials (clinicaltrials.gov NCT02951429, NCT02802345) that are currently enrolling participants. Nowadays, there is not an effective specific therapy for PH associated with IPF. Furthermore, several trials have demonstrated that pulmonary vasodilators might increase the *v*/*q* mismatch, which could result in a worsening of the hypoxemia and disease progression in IPF patients.

With the approval of pirfenidone and nintedanib, a new scenario is open for the role of these drugs in PH associated with IPF. A tyrosine kinase inhibitor as imatinib has been reported to improve exercise capacity and hemodynamics, but serious adverse events led to a high rate of study discontinuation in the active treatment arm, precluding definitive conclusions. The antiproliferative and anti-inflammatory activity of pirfenidone could theoretically be beneficial in the treatment of PH associated to IPF, but presently there is no definitive data that support this hypothesis. Thus, the possible beneficial role of nintedanib in PH due to IPF remains an open area of investigation [27]. 

Although hypoxemia correction confers a survival advantage in patients with COPD, the role of long-term oxygen therapy in PH-IPF has not been clearly established. Moreover, it has been suggested that long-term oxygen therapy for this disease does not result in either improvement of PAP or reversal of pulmonary vascular remodeling. Nevertheless, the only therapeutic approaches to PH in the setting of IPF currently recommended by the European Society of Cardiology/Eureopean Respiratory Society (ESC/ERS) guidelines are to correct hypoxia with supplemental oxygen therapy, to treat the underlying interstitial lung disease, and to refer the appropriate candidate for lung transplantation [34]. Pulmonary rehabilitation is another important measure to enhance the care of IPF patients with PH, since it improves exercise tolerance—as measured by the 6 MWT—and quality of life (Short Form-36 Healthy Survey). Lastly, in IPF patients with PH that is out-of-proportion to their degree of ventilatory restriction, referral to a PH specialist for diagnosis and treatment might provide some benefit. To the best of our knowledge, the role of regenerative therapies with stem cells have been assessed only in preclinical studies in experimental models of lung fibrosis or PH, but not in patients with PH-IPF. In the future, this modality of treatment might be a promising strategy in patients with IPF and underlying PH.

In conclusion, research in PH-IPF can be considered an emerging field. Further prospective studies are required to determine the best methods to detect PH in patients with IPF. In addition, and more importantly, current strategies for treating PH in IPF are far from perfect, and this fact should stimulate investigators to explore whether PH-IPF patients might benefit from novel therapies.

## 6. Sleep Disorders in Idiopathic Pulmonary Fibrosis

In healthy people, sleep is a state of restoration which covers approximately one-third of human life. Overall across studies, sleep is reported to be markedly disturbed in patients with IPF (Table 2). In fact, sleep-related disorders are increasingly recognized as important comorbidities in IPF patients. Sleep is associated with a degree of hypoventilation, especially during the vulnerable rapid eye movement (REM) period. This is normally well tolerated by healthy individuals, but it is potentially harmful for patients with chronic respiratory diseases. The possible lack of body and brain recovery during sleep in patients with IPF might be related to the negative consequences of the disease that take place both during wakefulness and during sleep time [44]. Thus far, research regarding sleep disorders associated to chronic respiratory disease has mainly focused on patients with COPD, but markedly fewer investigations have been performed in subjects with lung restriction, such as IPF.

Recent studies have revealed that moderate-to-severe obstructive sleep apnea (OSA) syndrome, defined by an apnea–hypopnea index (AHI) > 15 events per hour, is frequent in patients with IPF [45]. Moreover, although the increase of central respiratory drive in IPF is thought to be a protective factor for the development of sleep disorders, OSA has been recognized as an important, high-prevalence comorbidity in the latest official guidelines for the diagnosis and management of IPF. Two recent studies reported that OSA had a high incidence in patients with IPF [46,47]. Mermigkis et al. showed an incidence of 59% (44% mild and 15% moderate-severe OSA) in IPF cases [46]. Lancaster et al. showed similar results (88% had AHI ≥ 5 events per hour, 20% had mild OSA, and 68% had moderate-to-severe OSA) [47]. Both studies reported that the majority of the scored respiratory events were hypopneas. However, these studies are limited by the small number of patients included. Mermigkis et al. found a positive correlation between the AHI and body mass index values. They also found that REM AHI and total AHI were negatively correlated with FEV_1_ and FVC percentages. These findings suggest that obese IPF patients with decreased pulmonary function might have an increased risk for OSA, especially during REM sleep [45]. Mermigkis et al. found that TLC was inversely correlated with REM AHI. This finding might suggest a possible pathogenic link between IPF and OSA: upper airway instability could be aggravated by the decrease in lung volumes caused by IPF [46]. However, the pathogenic relationship, if any, between both diseases is probably complex. It is not clear whether OSA might appear during the natural course of the interstitial lung disease, as a consequence of lung function restriction, or whether it can promote gastroseophageal reflux disease (GERD) or increase oxidative lung stress through chronic nocturnal intermittent hypoxia, and these mechanisms can, by themselves, increase the risk for interstitial lung disease. An interesting study by Pillai et al. showed that 90% of patients with moderate IPF had GERD, 64% had a diagnosis of OSA, and 50% presented both diseases [48]. Investigators concluded that OSA is not a risk factor for GERD in IPF patients [48].

Anyway, most studies have agreed on the finding of disturbance of sleep architecture. The abnormal sleep macroarchitecture in patients with IPF is characterized by abnormal sleep stage distribution with reduced slow-wave and REM sleep, increased stage 1 sleep, multiple awakenings, decreased percentage of total sleep time, low sleep efficiency, and increased wake time after sleep onset. These patients also have an altered microarchitecture, as they present with an increased number of sleep microarousals resulting in sleep fragmentation. Therefore, these disorders in sleep architecture and the resulting poor quality of sleep impair the quality of life in patients with interstitial lung diseases (ILD) [49]. Another issue to take into account is that IPF patients present an altered sleep breathing pattern. Classic studies have described a distinct, high-frequency respiratory pattern in patients with interstitial lung diseases. Similarly, IPF patients exhibit an increased respiratory frequency during wakefulness that persists during sleep [46]. Finally, several studies have reported that IPF patients with the lowest awake oxygen saturation (SpO_2_) had greatest sleep-related desaturation, especially during REM sleep. Thus, nocturnal hypoxemia seems to be common between patients with IPF (SpO_2_ < 90% was present during more than one-third of the total sleep time), and this also has a negative impact on the quality of life and daytime function [44]. These findings open the door for the potential use of overnight oxygen supplementation as a palliative treatment of IPF. 

There is also some evidence that REM-associated sleep-related breathing disorders (SRBD) in IPF patients might play an important role, not only in the quality of sleep and life, but also in the mortality associated with ILD. The sleep-related desaturation in these patients might contribute to pulmonary vascular disease development, with PH as one of the possible consequences. Therefore, it might be a risk factor for increased mortality. It must be noted that, in the study by Corte et al., nocturnal oximetry was not able to distinguish those patients with ILD who desaturated because of concomitant OSA [50]. Hence, more complex studies, like nocturnal respiratory poligraphy, might be indicated to evaluate sleep disorders in IPF, in order to establish the optimal treatment in these cases. 

Since SRBD in IPF patients seem to be so prevalent, even in the initial phase of IPF, and are likely to be underdiagnosed, it is important for physicians to recognize the clinical phenotype of these patients. Early detection and referral to a sleep center for evaluation is essential to promptly initiate treatment. The therapy for SRBD in patients with IPF is decided on an individual basis, although the optimization of sleep and quality of life through SRBD treatment should be recognized as a primary goal [49].

In this sense, available tools for treating patients with IPF and underlying SRBD are oxygen therapy for REM-related SRBD and continuous positive airway pressure (CPAP) for the overlap of IPF and OSA. Oxygen administration is reasonably well tolerated and recommended for advanced forms of IPF. Nevertheless, long-term home oxygen therapy in IPF patients did not improve survival when respiratory failure had been established. Moreover, only one study was carried out to analyze the use of CPAP in patients with IPF and OSA. This investigation found that these patients had a significant improvement in quality of life and sleep, daily living activities, and survival [49]. The study provides the first evidence that treatment of some comorbidities such as OSA might influence mortality in IPF patients. However, further research is needed to assess whether OSA contributes to IPF progression and whether CPAP treatment might influence this progression. Also, the impact on quality of life of untreated OSA in these patients should be adequately studied. Finally, one of the most interesting future research fields is to study the potential effect of the new antifibrotic therapies on the underlying comorbidities, such as the development of OSA.

In conclusion, alterations in sleep architecture, breathing pattern, and desaturations in IPF patients, especially during REM sleep, as well as coexisting OSA, might have a significant negative impact on the sleep quality of these patients. Physicians should ask the patient for symptoms during the night and should maintain a high degree of clinical suspicion for the presence of possible comorbid sleep disorders. As IPF is a progressive and life-threatening disease, the role of the poor quality of sleep as a significant comorbidity must be reconsidered. The effective treatment of comorbidities in IPF patients, such as sleep disorders, should be a field of future research.

## 7. Acute Exacerbation

Idiopathic pulmonary fibrosis is a progressive disease that can present different types of evolution overtime. Some patients may progress slowly, while others experience a rapid progression, with fast decline of the lung function. Other subjects, however, may suffer episodes of sudden deterioration. These episodes are defined as acute exacerbations (AE-IPF) [51].

Traditionally, AE-IPF have been defined as acute, clinically significant episodes of respiratory deterioration without an identifiable cause. In recent years, the International Working Group in AE-IPF has reviewed this definition as well as its diagnostic criteria (Table 3). The last workshop eliminated the concept that the respiratory event should be necessarily idiopathic, as stated in previous definitions of AE-IPF. Also, the strict 30-day time period interval has been made more flexible. It is no longer necessary to demonstrate the absence of an infectious cause by endotracheal aspirate or bronchoalveolar lavage, and there may be signs of heart failure, although this condition should not fully explain the severity of the respiratory deterioration [52].

The incidence of these episodes varies between 2% and 16% per year. Results from clinical trials of nintedanib and pirfenidone suggest that IPF therapies may help to prevent the development of AE-IPF [52].

The aetiology of AE-IPF remains unknown. It is thought to have similarities with the aetiology of acute lung injury (ALI), so that experimental models of ALI have been used for the study of AE-IPF. AE-IPF is most likely triggered by an acute event that leads to widespread acute lung injury [53]. There is evidence of viral infections in the AE-IPF, both microbiological (genomic detection) and supported by post-mortem studies. There are additional data provided by epidemiological studies. Episodes of AE-IPF are more frequent in wintertime and in those patients who receive immunosuppressive medication [54]. Other potential triggers that have been proposed are GER, pulmonary surgery (either lung cancer resection or a lung biopsy), and the performance of bronchoalveolar lavage [55].

AE-IPF is more common in patients with physiologically and functionally advanced disease (low FVC, DLCO, 6 MWT distance, and poor baseline oxygenation). Other identified risk factors are younger age, coronary artery disease, and high body mass index [56].

The presence of AE-IPF not only implies a high mortality during the event (up to 50%), but also a significant decrease in short-term survival after the episode. IPF patients who suffer an exacerbation have a subsequent average survival of three to four months. Mortality reaches 90% in cases in which the patient needs mechanical ventilation [57].

Given the lack of knowledge on the pathogenesis of AE-IPF, it is understandable that there is no current effective therapy available. Treatment with corticosteroids is a recommendation based on weak-strength, very low-quality evidence. The evidence available is based on isolated case reports, and there are no controlled clinical trials that endorse the benefits of their use [58]. The exact dose, route of administration and duration of the treatment are not precisely known.

The use of mechanical ventilation is also controversial because of the high mortality observed when this therapy is employed, so it is usually reserved as a life-supporting procedure for those patients who will undergo a lung transplantation. Anyway, the decision to initiate mechanical ventilation must be carefully considered on an individual basis [58].

Several potential therapies for AE-IPF have been described over the last decade, but these studies are mostly small and uncontrolled: cyclosporine, rituximab, tacrolimus, azithromycin, and intravenous thrombomodulin are some examples [51].

## Figures and Tables

**Figure 1 medsci-06-00071-f001:**
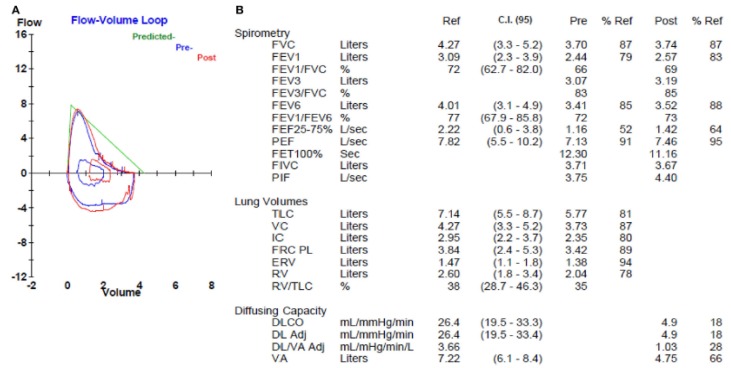
Pulmonary function test in a patient with combined pulmonary fibrosis and emphysema. The flow-volume loop (**A**) shows a mild obstructive morphology but normal spirometry (**B**). Lung volumes (**B**) are normal. The diffusing capacity (DL) (**B**) is markedly reduced. C.I. (95): 95% confidence interval; FVC: forced vital capacity; FEV_1_: forced expiratory volume in the first second; FEF: forced expiratory flow; PEF: peak expiratory flow; FET: forced expiratory time; FIVC: forced inspiratory vital capacity; PIF: peak inspiratory flow; TLC: total lung capacity; VC: vital capacity; IC: inspiratory capacity; FRC: functional residual capacity; ERV: expiratory reserve volume; RV: residual volume; DLCO: diffusing capacity of carbon monoxide; VA: volume adjusted.

**Figure 2 medsci-06-00071-f002:**
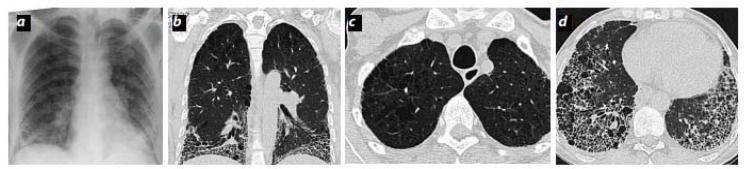
Combined pulmonary fibrosis and emphysema (CPFE). (**a**) Bilateral chest radiography with basal reticular opacities. Increased diameter of the interlobar artery (pulmonary hypertension). (**b**–**d**) High-resolution computed tomography (HRCT) axial slices (apexes and bases) and coronal reconstruction. Areas of low centrilobular density without defined wall with apical predominance (centrilobular emphysema). Honeycombing is basal, bilateral, and symmetrical with loss of volume of the lower lobes. Traction bronchiectasis is observed.

**Table 1 medsci-06-00071-t001:** Hemodynamic classification of pulmonary hypertension (PH) due to lung disease.

Terminology	Haemodynamics (Right Heart Catetherization)
**COPD/IPF/CPFE without PH**	mPAP < 25 mmHg
**COPD/IPF/CPFE with PH**	mPAP ≥ 25 mmHg
**COPD/IPF/CPFE with severe PH**	mPAP > 35 mmHg, or mPAP ≥ 25 mmHg in the presence of low cardiac output (CI < 2.5 L/min/m^2^, not explained by other causes)

Adapted from Galiè et al. and Harari et al. [29,36]. COPD: chronic obstructive pulmonary disease; CPFE: combined pulmonary fibrosis and emphysema; IPF: idiopathic pulmonary fibrosis; CI: cardiac index; mPAP: mean pulmonary arterial pressure.

**Table 2 medsci-06-00071-t002:** Sleep disorders in IPF.

Sleep Macro and Microarchitecture	Increased Stage N1 Sleep, Arousal Index, WASO Decreased REM, Slow-Wave Sleep, and Sleep Efficiency
Respiratory pattern	Increased respiratory frequency during sleep
	Rapid and shallow breathing (especially during REM sleep)
Nocturnal oxygenation parameters	Episodic desaturation during REM sleep
	Desaturation during NREM sleep
	Desaturation due to respiratory events (apneas and hypopneas)
Sleep-disordered breathing and other sleep problems	Increased incidence of OSA
	Increased periodic leg movements during sleep Insomnia
	Nocturnal cough

Adapted from Mermigkis, et al. [44]. N1: stage 1; WASO: wake time after sleep onset; REM: rapid eye movement; NREM: non-REM; OSA: obstructive sleep apnea.

**Table 3 medsci-06-00071-t003:** Definition and diagnostic criteria for acute exacerbation of IPF.

**Revised Definition**
Acute, clinically significant respiratory deterioration characterized by evidence of new widespread alveolar abnormality
**Revised Diagnostic Criteria**
Previous or concurrent diagnosis of IPF
Acute worsening of development of dyspnea typically with less than one-month duration
CT scan with new bilateral ground-glass opacity and/or consolidation superimposed on a background with usual interstitial pneumonia (UIP) pattern
Deterioration not fully explained by cardiac failure or fluid overload

Adapted from Acute Exacerbations of IPF, an International Working Group Report [51].

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
