# Peer review of "Comorbidities and Complications in Idiopathic Pulmonary Fibrosis"

_medsci, 2018, doi:10.3390/medsci6030071_

Reviewer 1 Report

It is a very interesting paper, I have some comments:

The start of the paper could have an introduction and no start with the emphysema directly, maybe explain the importance of the comorbidities in IPF. I suggest Respir Med. 2017 Aug;129: 46-52.  

In the treatment of CPFE what is the recommendation about the fibrosis or only treat the emphysema? (there is no a lot of information but it could be interesting in this paper)

Author Response

According to the reviewer's suggestions, two paragraphs have been added to the text. One is the introduction and the other is the CPFE treatment.

Reviewer 2 Report

This is a review on co-morbidities associated with idiopathic pulmonary fibrosis. It contains a complete description of the common complications of the fibrotic disease. The main problem is the English language. The manuscript contains many grammatical and syntax errors that make difficult to follow its content. A native speaker and expert-in-the-field person must carefully revise the manuscript. 

Author Response

According to the author's suggestions, a revision of the grammar and syntax of the manuscript has been carried out